# Aspects of Self-Organization of Tribological Stressed Lubricating Greases

**Erik Kuhn** 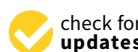

TREC, Institute of Engineering Design and Product Development, Hamburg University of Applied Sciences, Berliner Tor 21, 20099 Hamburg, Germany; erik.kuhn@haw-hamburg.de; Tel.: +49-4042875-8623

**Abstract:** Lubricating greases are markedly visco-elastic materials. Stressed by a friction process, this special material shows a drop of a measured shear stress or viscosity. This typical behaviour is observed in a number of papers and, therefore, is well known. Some different explanations can be found but most of them describe a structural degradation caused by the friction process. This paper attempts to elucidate the conditions that promote that structural change and understand this behaviour as an intrinsic response of the system.

**Keywords:** lubricating grease; structural degradation; self-organization

## 1. Introduction

Tribology deals with defined systems which are stressed by a friction process. As an observer of that process, tribologists can record the reaction of the system in different ways and can try to interpret the obtained process behaviour.

It may be interesting to look at the energetic situation and follow the course of the energetic state, because it seems to be important to determine the driving forces of the observed process. These fundamental behaviours and important questions are described, for example, in publications like [1–5].

In order to derive the observed subsystem from the general tribo system, a volume element of the grease film is modeled and observed. The general behavior of tribo systems is transferred to the subsystem in order to gain new insights into the origin of the special lubricant behviour. It is an attempt to get a better understanding of the intrinsic reaction of a tribological stressed grease.

## 2. Fundamental Definitions

In view of the presented work, the general understanding of friction and wear should be oriented to the principle system behaviour. That means we are observing the effort of a tribosystem that has become non-stable to, once again, reach a stable situation. It follows that "friction" is the introduction of mechanical energy into a tribosystem and "wear" is the dissipation of that energy with a concurrent increase in entropy. It's a question of defintion but here we find that friction is the introduction of energy into the system and the reaction is to mitigate the disturbance with a concurrent increase in entropy. This does not contradict traditional definitions but allows to subsume all tribological processes to the general terms of friction and wear.

Using this concept and following the principle of Le Chatelier–Prigogine Figure 1. was created to illustrate the given definitions.

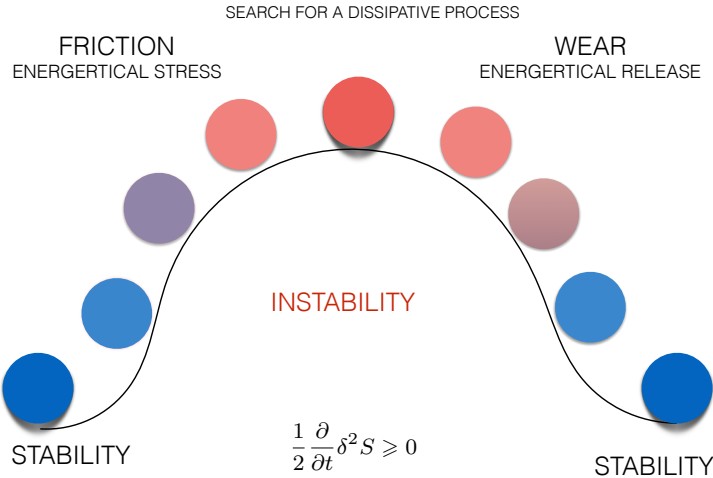

**Figure 1.** Illustration to the general behaviour of a tribo system [6].

## 3. The General Grease Behaviour

A number of previous experimental works [7–9] show the influence of shear rate, stress, temperature, type of base oil, type of solid (thickener) and thickener content on the curve behaviour (for example $\tau$ vs. $\dot{\gamma}$). Papers from [9–14] are focused on the structure and present the system behaviour, as influenced by the observed grease structure.

Most of the mentioned experimental works on greases were done by the use of a rheometer. Both rheometer modes are useful—the rotational mode and the oscillating mode. While we are far removed from process conditions for real machine elements, the very high sensitivity of a rheometer makes this device an important experimental tool for our research topic. However, we must recognise that all rheometer tests are friction tests. The type of friction that occurs during a shear process of a grease in a rheometer can be identified as inner-friction (or liquid-friction, or lubricant-friction). Even though we know what occurs in the background and influences the curve behaviour that we observe—the measured data are energy expenditures. That means, for wear investigations, rheometer tests can only involve non-direct measurements.

The idea of two phases of degradation was given, for example, in [7,15]. First, a strong dependence on time with decreasing shear stress and second, a state near an equilibrium. That means, in the second part, we assume there to be a balance between degradation and coagulation inside the grease film. In [16], we find evidence of a so-called running-in period followed by a stationary degradation phase. Later, Zhou et al. [17] reported the existence of two different slopes of a degradation curve from another point of view.

This first part of the grease behaviour vs. stress time is interesting because we are able to observe the system reaction. During the second part, the stationary period, we can only assume what mechanisms occur. In recent years, the general investigations of friction and wear were transferred to the special material and lubricant—lubricating grease. This means that, in addition to the friction investigations, irreversible structural changes, as an intrinsic reaction, are also of interest [6,11,18–21].

## 4. A Thermodynamic Point of View

We deduce a sub system from an ordinary tribo system. This sub system consists of a stressed grease volume element in relative motion to another volume element. To investigate the stressed grease, an open thermodynamic system is defined, whereby we observe an exchange of matter and heat through the system boundary and we observe several processes inside the system.

According to Figure 2, it is well known that after starting a tribological process and running through a time-dependent period, the grease behaviour enters a stationary regime. That means the

solid inside the grease film forms a more or less stable structure. Observations of the behaviour can be viewed as evidence of self-organization. Some interesting self-organization effects are listed in [1] and extended in Table 1. As we know from [22], some thermodynamic systems that operate far from equilibrium and that are open systems lead to an increase in orderliness and self-organization. Furthermore, in [23], the information that friction and wear can lead not only to deterioration but to self-organization was reported. As mentioned in [24], self-organization leads to the formation of dissipative structures.

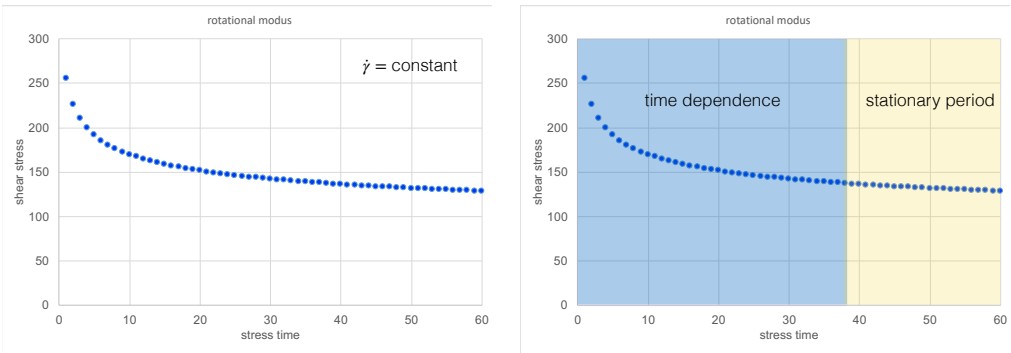

**Figure 2.** General grease behaviour in a rheometer experiment, shown as shear stress vs. stress time with a constant shear rate.

**Table 1.** Effects of self-organization.

| Effect | Driving Force | Condition to Initiate | Final Configuration |
| --- | --- | --- | --- |
| stationary microtopography after running in | feedback due to coupling of friction and wear | wear affects microtopography until it reaches the stationary value | minimum friction and wear [1] |
| in situ tribofilm formation | chemical reaction leads to film growth | wear decreases with increasing film thickness | minimum friction and wear at stationary film thickness [1] |
| stationary grease structure after time dependence | involved fragmentation and coagulation | thermodynamic criteria | minimum entropy production |

Reference [25] argue that self-optimization can be initiated in a system after it has lost its thermodynamic stability.

To obtain information about the grease behaviour during the first stress period, it is necessary to investigate the criterion for stability. If instability occurs, the process of self-optimization can start.

## 5. Criterion for Instability

As a condition for an appearance of stability can be written [22]

$$\frac{1}{2}\frac{\partial}{\partial t}(\delta^2 S) > 0 \tag{1}$$

it presents the second variation of entropy production per time.

The time deviation for $\delta^2 S$ is related to the entropy production by disturbance [26] and from [22], we know

$$\frac{1}{2}\frac{\partial}{\partial t}(\delta^2 S) = \sum X_h \cdot J_h \tag{2}$$

with $X_h$ as a thermodynamic force that induces a heat flux and $J_h$ as the heat flux. From [27], Equations (3) and (4) are used

$$J_h = -\lambda grad T \tag{3}$$

$$X_h = \frac{grad T}{T^2} \tag{4}$$

for the investigated general friction process, it is proposed

$$J_h = \tau \cdot \dot{\gamma} \tag{5}$$

The entropy variation of the observed grease volume element $dS$ is composed of the following components

$$dS = dS_i + dS_e \tag{6}$$

where $dS_i$ is the change of entropy production and $dS_e$ is the change of the entropy flow. This investigation considers a system that consists of one rubbing body—the stressed grease volume element. In general, the entropy production can vary due to any process in the stressed body. We obtain the following Equation for the entropy production rate

$$\frac{dS_i}{dt} = X_h \cdot J_h = \frac{(\tau \cdot \dot{\gamma})^2}{-\lambda \cdot T^2} \tag{7}$$

As mentioned before, a self-optimization process starts in system after a loss in thermodynamic stability. The task is now to find conditions that increase the probability to start a self-optimization process. We must investigate the criterion for stability

With Equation (2), we have

$$\frac{1}{2} \frac{\partial}{\partial t}(\delta^2 S) = \delta\left(\frac{\tau \cdot \dot{\gamma}}{-\lambda \cdot T^2}\right)\delta(\tau \cdot \dot{\gamma}) \tag{8}$$

According to [25], let us introduce a parameter $\varepsilon$ to describe the distance from the equilibrium. In our case, $\varepsilon$ expresses the difference between the grease structure in an equilibrium and the structure in an a stationary non-equlibrium state. We use the dependence $\tau(\varepsilon)$ and $\lambda(\varepsilon)$. The following is obtained

$$\frac{1}{2} \frac{\partial}{\partial t}(\delta^2 S) = \delta\left[\frac{\tau \cdot \dot{\gamma}}{-\lambda \cdot T^2}\right]\delta(\tau \cdot \dot{\gamma}) = \frac{\dot{\gamma}^2}{T^2}\left[\frac{1}{\lambda}\left(\left(\frac{\delta\tau}{\delta\varepsilon}\right)^2 - \frac{\tau}{\lambda^2}\frac{\delta\tau}{\delta\varepsilon}\frac{\delta\lambda}{\delta\varepsilon}\right)\delta\varepsilon^2\right] \tag{9}$$

The right side of Equation (9) must become negative if instability is to arise. That means an increase or decrease of shear stress $\tau$ and heat conductivity $\lambda$ simultaneously with $\varepsilon$ increments.

$$\frac{\delta\tau}{\delta\varepsilon}\frac{\delta\lambda}{\delta\varepsilon} > 0 \tag{10}$$

From rheological experiments, see Figure 2, we know that there is a decreasing shear stress for a constant shear rate if wear happens. The same process behaviour can be assumed for heat conductivity. The tribological process produces more small particles covered by an oil coat.

Now, the general process should be divided in more detail. Three mechanisms should be selected. These are

- shear process of the base oil
- the process of fragmentation of the solid agglomerates
- process of coagulation or agglomeration caused by collision of particles.

That is

$$\frac{dS_i}{dt} = \frac{dS_{i(\text{OIL})}}{dt} + \frac{dS_{i(\text{FRAG})}}{dt} + \frac{dS_{i(\text{COAG})}}{dt} \tag{11}$$

For the fragmentation process, the energy to overcome a critical deformation determines the entropy production. For the agglomeration caused by collision of the particles, the kinetic energy of the flowing agglomerates influences the entropy production.

$$\frac{dS_i}{dt} = \frac{(\tau_{(\text{OIL})} \cdot \dot{\gamma})^2}{-\lambda \cdot T} + \frac{\gamma_{critic}^2 \cdot G' \cos\delta^{-1} \cdot F_f}{T} + \frac{[e_{kin(b)} - (e_{kin(a)} + e_{def})] \cdot k_C}{T} \tag{12}$$

Here, $\gamma_{critic}$ is the deformation of the transition from an elastic to plastic state, $G'$ is the storage modulus from oscillating measurement, $cos\delta^{-1}$ from $tan\delta$ is the loss factor, $F_f$ is the agglomeration rate, $e_{kin(b)}$ is the kinetic energy before collision, $e_{kin(a)}$ is the kinetic energy after collision ($e_{kin(a)} = 0$ for agglomeration), $e_{def}$ is the energy that deforms the particle, and $k_C$ is the collision rate.

This leads to

$$\sum_i \delta X_i \cdot \delta J_i = -\frac{1}{\lambda \cdot T^2} \left[ \cdot \left( \frac{\delta \tau_{(\text{OIL})}}{\delta \epsilon} \dot{\gamma} + \tau \frac{\delta \dot{\gamma}}{\delta \epsilon} \right) \delta \epsilon \right]^2 + \frac{G' \cos\delta^{-1}}{T} \left[ 2\gamma_{critc} \frac{\delta \gamma_{critic}}{\delta \epsilon} \frac{\delta F_f}{\delta \epsilon} (\delta \epsilon)^2 \right] +$$
$$\frac{1}{T} \left[ \frac{\delta e_{kinB}}{\delta \epsilon} \frac{\delta k_L}{\delta \epsilon} (\delta \epsilon)^2 \right] \tag{13}$$

We have three addends at the right-hand side of Equation (13) and must look for a negative entropy production rate to determine a loss of thermodynamic stability.

- the first addend is always negative because of the sign and the square
- in order for the second addend to become negative, it would be necessary to meet the following assumption

$$\frac{\delta \gamma_{critic}}{\delta \epsilon} < 0; \frac{\delta F_f}{\delta \epsilon} > 0 \quad \text{and vice versa} \tag{14}$$

- and for the third addend to become negative

$$\frac{\delta e_{kinB}}{\delta \epsilon} < 0; \frac{\delta k_L}{\delta \epsilon} > 0 \quad \text{and vice versa} \tag{15}$$

Of course, if all addends are negative, we have the largest probability for an initialization of a self-optimization process. Let us focus the investigation on the second addend.

For example, we need a decreasing critical deformation $\gamma_{critic}$ and an increasing fragmentation rate $F_f$.

## 6. Experimental Work

From the experimental work of [28], we can obtain some information of the grease behaviour during an increasing tribological stress. An amplitude sweep of an oscillating rheometer measurement delivers the possibility to obtain information about the transition from elastic range to plastic range. The end of the so-called linear visco-elastic range (LVE) can be quantified by the value of the critical deformation $\gamma_{critic}$.

This critical deformation is influenced by the content of solid in a lubricating grease. The observation of [28] shows a decreasing value of the critical deformation with an increasing content of solid for the used greases. The physical bonds between the solid fibres are much weaker compared with the bond inside the base oil. Therefore, the result in Figure 3 could be expected.

The interpretation from this experimental result and the correlation with the investigation of the process stability lead to the assumption of an earlier or faster self-optimization process for higher contents of solid in the lubricating grease. However, we must note that we investigated the probability of self-optimization.

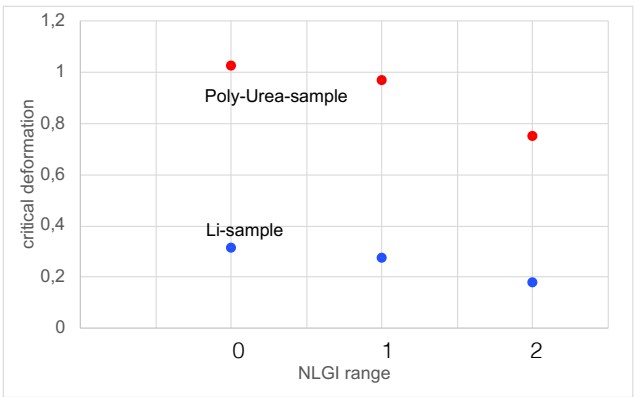

**Figure 3.** Critical deformation $\gamma_{critic}$ vs. NLGI (National Lubricating Grease Institute)-grade as reported in [28].

Finally, rheometer tests were performed with two grease samples, with a Li-soap with NLGI grade 1 and 2. These grease samples contained only a Li-soap and a base oil. The tests were made with a plate-plate system for constant shear rate $\dot{\gamma}$ and constant temperature $\vartheta$. The curves in Figure 4. are the mean values of 5 tests with the same conditions. The standard deviation was less than 5%.

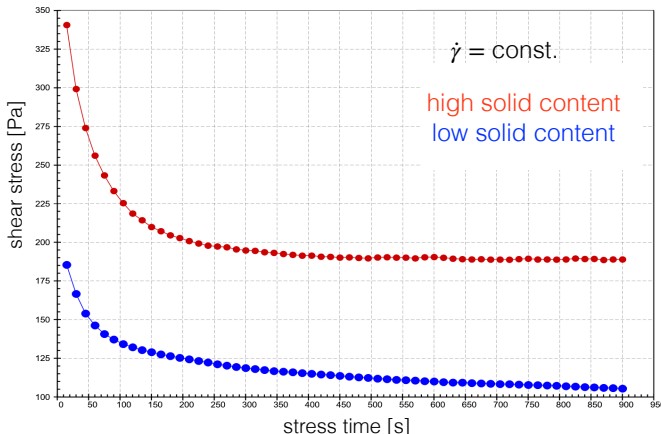

**Figure 4.** Non-stationary period and stationary period of the two grease samples with different solid contents.

In the light of this, a different behaviour for the transition between the time-dependent period and stationary period can be observed.

## 7. Some Thoughts About the Stationary Period

We assume a constant shear stress over a stress time and observe the process behaviour in a rheometer experiment. After a certain time, the slope of a $\tau$ vs. $t$-curve is nearly constant. This period is called the stationary period and can be characterized as a stationary Non-Equilibrium. Following different researchers, such as R. Czarny [7] and J. M. Franco (personal communication), two main processes are assumed. These are the fragmentation of the agglomerates and the agglomeration (coagulation) caused by particles collision. It seems that there exists a balance between theses processes during the stationary period.

Using the terminology of Nosonovsky and Bushan [29], the fragmentation and agglomeration should be characterized as degradation and healing processes. Two processes are considered: One process, characterized by a degradation parameter $\xi$, and another process, characterized by

a healing parameter $\zeta$. The fragmentation of the lubricating grease structure is connected with $\xi$ and the agglomeration is connected with $\zeta$.

As known from non-equilibrium thermodynamics, the thermodynamic flows are connected with the thermodynamic forces via

$$J_k = \sum_i L_{ki} \cdot X_i \tag{16}$$

here, $L_{ki}$ is the Onsager coefficient based on the Onsager reciprocity relationship $L_{ki} = L_{ik}$ [27]. Degradation forces and healing forces are external forces and the flows are related to the forces. It can be obtained from [29]

$$J_{deg} = K \cdot X_{deg} + M \cdot X_{heal} \tag{17}$$

$$J_{heal} = M \cdot X_{deg} + H \cdot X_{heal} \tag{18}$$

where $K, M, H$ are the Onsager coefficients. $J_{deg}$ stands for $J_{frag}$ and $J_{heal}$ stands for $J_{coag}$. With the idea that degradation and healing grow when positive forces are applied and degradation and healing decrease when opposite forces are applied, [29] show that $K > 0, H > 0$ and $M < 0$.

Furthermore, [29] presents a model of self healing and define some assumptions. For example, a constant degradation force with $J_{deg} = \varrho$ and a healing force that is proportional to the healing parameter $J_{heal} = f(\xi)\zeta$. The dependence of the healing rate on the structural change or fragmented particles is described with the function $f(\xi)\zeta$. This is used for the investigation in this paper because a certain amount of the energy during collision is dissipated in the oil coat of the particles. According to the literature, a constant dependence $f(\xi) = -\omega$ is assumed in the first step.

Now, a balance between degradation and healing or better, between fragmentation and agglomeration, is considered. It is obtained as

$$\dot{\xi} = -K\omega\zeta - M\omega\zeta \tag{19}$$

$$\dot{\zeta} = -M\omega\zeta - H\omega\zeta$$

The solution of the above system of differential equations delivers

$$\xi = a \cdot e^{-\omega(M+H)t} + b \tag{20}$$

$$\zeta = a \cdot \frac{M+H}{K+M} \cdot e^{-\omega(M+H)t}$$

where $K$ corresponds to degradation and $H$ corresponds to healing. If the condition is $K = H$, the functions are identical but shifted by b. It seems logical because the fragmentation or structural level will never reach the base oil level.

## 8. Conclusions

A liquid friction contact for a lubricating grease is investigated as an open thermodynamic system. An exchange of matter and heat through the system boundaries and some mechanisms inside are considered. The tribological stress pushed the observed volume element out of the thermodynamic equilibrium. That means the system lost its stability.

Upon observation of rheometer tests during a shear experiment with constant shear rate, a drop in shear stress vs. stress time can be observed. This behaviour should be interpreted as the system reacting in order to regain stability. This is an intrinsic behaviour of the tribo-thermodynamic system.

The investigation of the stability criteria for the observed topic delivers information about the driving forces of the tribo-process. Furthermore, it shows the influence of selected parameters such as the solid content on the system behaviour to become stable.

Finally, an analytical description is presented to obtain more information about the degradation and agglomeration during the stationary period of a shear process.

**Acknowledgments:** The work was supported by German Ministry of Research and Education.

**Conflicts of Interest:** The author declare no conflict of interest.

## Abbreviations

| | |
|---|---|
| G′ | storage modulus [Pa] |
| S | entropy [J/K] |
| T | temperature [K] |
| a,b | constants of integration |
| e | energy density [J/m$^3$] |
| $\gamma$ | deformation [-] |
| $\dot{\gamma}$ | shear rate [1/s] |
| $\lambda$ | heat conductivity [W/mK] |
| $\tau$ | shear stress [Pa] |

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
