# Peer review of "Aspects of Self-Organization of Tribological Stressed Lubricating Greases"

_lubricants, doi:10.3390/lubricants8030028_

Round 1

Reviewer 1 Report

The work is interesting and valuable. I recommend the article for printing in the journal Lubricants.

Author Response

Thank You very much for your critical revision of the manuscript and for your recommendation

Reviewer 2 Report

The following points should be addressed:

  1. Abstract: You mention that 'This typical behaviour is observed in a number papers'. Add references.
  2. The introduction is rather short. The purpose of this work and the state-of-the-art that it brings should be better highlighted. 
  3. Throughout the manuscript the references are not marked [?].
  4. Throughout the manuscript the figures are not marked ??
  5. An Annex should be added to explain all symbols.
  6. The experimental tests should be better described.
  7. Axes in figure 4 are not described. In addition, greases are quite inhomogeneous materials. What is the spread of experimental values?
  8. In tribological measurements the material characteristics (composition, surface topography etc.) as well as the contact conditions (load, speed etc.) have a strong effect on the performance of the tribosystem. Do you consider this is your model?  

Author Response

Thank you very much for your critical note and your support.

I like to answer point by point

  1. It is not possible to add references inside the abstract. This journal do not allow this.
  2. I extend the introduction with this text passage:     In order to derive the observed subsystem from the general tribo-system, a volume element of the grease film is modeled and observed. The general behavior of tribo- systems is transferred to the subsystem in order to gain new insights into the origin of the special lubricant behaviour. Its the attempt to get a better understanding of the intrinsic reaction of a tribological stressed grease.
  3. I hope I understand right. In the reference list are no brackets like [1]. But this comes from the template of the journal!
  4. The figures and tables are without brackets in the captions (template from the journal). I delete the brackets when I mentioned it in the text.
  5. I have explained every symbol in the text. But now I have add an additional abbreviation with some fundamental symbols I used.
  6. For the experiments I did by my self (not from literature) I add the following sentences:The tests were made with a plate-plate-system for constant shear rate γ ̇ and constant temperature θ. The curves in Figure 4. are the mean values of 5 tests with the same conditions. The standard deviation was less than 5%.  
  7. I replaced the figure. Now the axis are described. In the old version the names of the axis are situated above the diagram.
  8. You are right. In a grease lubricated contact the material and contact conditions influence the performance of the system. But here I observe the lubricant. I try to look inside the lubricant (fluid friction) and the stress is shearing the grease in a rheometer. We have no influence by the plate material or by pressure. Effects like in real (machine) contact are not considered. Experiments were done with a rheometer.